# Fenestrated Endothelial Cells across Organs: Insights into Kidney Function and Disease

**DOI:** 10.3390/ijms25169107

**Published:** 2024-08-22

**Authors:** Xingrui Mou, Sophia M. Leeman, Yasmin Roye, Carmen Miller, Samira Musah

**Affiliations:** 1Department of Biomedical Engineering, Pratt School of Engineering, Duke University, Durham, NC 27710, USA; 2Department of Computer Science, Duke University, Durham, NC 27710, USA; 3Department of Biology, Duke University, Durham, NC 27710, USA; 4Center for Biomolecular and Tissue Engineering, Duke University, Durham, NC 27710, USA; 5Division of Nephrology, Department of Medicine, School of Medicine, Duke University, Durham, NC 27710, USA; 6Department of Cell Biology, Duke University, Durham, NC 27710, USA; 7Faculty of the Developmental and Stem Cell Biology Program, Duke Regeneration Center, Duke MEDx Initiative, Duke University, Durham, NC 27710, USA

**Keywords:** endothelial cells, fenestrations, glomerulus, kidney disease, disease models, microphysiological systems, vasculature

## Abstract

In the human body, the vascular system plays an indispensable role in maintaining homeostasis by supplying oxygen and nutrients to cells and organs and facilitating the removal of metabolic waste and toxins. Blood vessels—the key constituents of the vascular system—are composed of a layer of endothelial cells on their luminal surface. In most organs, tightly packed endothelial cells serve as a barrier separating blood and lymph from surrounding tissues. Intriguingly, endothelial cells in some tissues and organs (e.g., choroid plexus, liver sinusoids, small intestines, and kidney glomerulus) form transcellular pores called fenestrations that facilitate molecular and ionic transport across the vasculature and mediate immune responses through leukocyte transmigration. However, the development and unique functions of endothelial cell fenestrations across organs are yet to be fully uncovered. This review article provides an overview of fenestrated endothelial cells in multiple organs. We describe their development and organ-specific roles, with expanded discussions on their contributions to glomerular health and disease. We extend these discussions to highlight the dynamic changes in endothelial cell fenestrations in diabetic nephropathy, focal segmental glomerulosclerosis, Alport syndrome, and preeclampsia, and how these unique cellular features could be targeted for therapeutic development. Finally, we discuss emerging technologies for in vitro modeling of biological systems, and their relevance for advancing the current understanding of endothelial cell fenestrations in health and disease.

## 1. Introduction

Organ-specific endothelial cells (ECs) have distinct morphological properties and expression patterns of gene networks that allow them to regulate signals in their microenvironment and function in tissue and organ development, homeostasis, and regeneration [1,2,3]. For example, the network of capillaries in the central nervous system consists of endothelial cells interconnected by tight junctions to form the blood–brain barrier (BBB) that selectively prevents some molecules, drugs, and pathogens from entering the brain [4]. Conversely, sinusoidal ECs—primarily located in the liver, spleen, bone marrow, and some endocrine glands—lack an organized basement membrane but possess open pores called fenestrations and intercellular gaps. These fenestrations enable the free exchange of water and provide a conduit for large solutes, including otherwise impermeable components of plasma [4,5]. ECs can also develop fenestrations through the invagination of the plasma membrane to produce transcellular pores (50 to 300 nm diameter) that facilitate the exchange of small-molecule nutrients and waste between the vasculature and surrounding tissues. This type of fenestration is critical to organs such as the kidney, which filters one liter of blood—approximately 20% of the blood pumped by the heart—per minute and performs essential functions including the removal of waste and toxins from the bloodstream and regulation of blood pressure. The fenestrated ECs facilitate the high rate of exchange between the intra- and extra-vascular compartments of the nephron [6]. In some pathological kidney conditions, glomerular ECs lose their fenestrations, leading to progressive kidney degeneration and end-stage kidney disease (ESKD). Despite well-known molecular associations with the induction and maintenance of glomerular EC fenestrations, particularly plasmalemma vesicle-associated protein (PLVAP) and vascular endothelial growth factor A (VEGFA), respectively, the role of fenestrations in kidney tissue development and (patho)physiology remains elusive. Below, we discuss the development and function of endothelial cell fenestrations in multiple organs, with emphasis on kidney homeostasis and dysfunction. In addition, this review article leans into the current and prospective uses of microphysiological models to advance research on this topic. Moreover, we provide insights into current knowledge gaps and in vitro models coupled with human stem cell technologies, which could help improve understanding of fenestration development, mechanisms, and relevance to disease onset and progression.

## 2. Endothelial Cell Fenestrations: From Development to Structure and Function

### 2.1. Structure and Function

Endothelial fenestrations are transcellular pores of 50 to 300 nm diameter across the plasma membrane that are organized within multiple sieve plates separated by cytoplasmic ridges [7,8]. The number and diameter of endothelial fenestrations are dynamically regulated by the local microenvironment. External stimuli, including inflammation, dietary fat load, circulating vasoactive cytokines, and hormones, can change the distribution of fenestrations [9]. For example, during intestinal inflammation, ECs in the brain’s choroid plexus downregulate PLVAP, a type II transmembrane glycoprotein that is presented on caveolae, trans-endothelial channels, and fenestrations of ECs—which causes the closure of fenestrations and leads to a more selective barrier that modulates immune cell translocation into the cerebrospinal fluid [10]. Fenestrated ECs are subdivided into three categories based on their location and structural features: diaphragmed and fenestrated, non-diaphragmed and fenestrated, and non-diaphragmed “discontinuous” cells (Figure 1A). Diaphragmed and fenestrated cells are characterized by transcellular pores organized by thin fibril diaphragms spanning from the center of the pores [11], like the spokes on bicycle wheels. This cell type includes gastrointestinal ECs and peritubular ECs, which require high water permeability in a controlled manner. Non-diaphragmed and fenestrated ECs are characterized by open transcellular pores without the presence of diaphragms. Glomerular ECs (GECs) are one example of this cell type, containing approximately 60 to 80 nm diameter fenestrations that occupy around 50% of the cell surface area [8]. Due to the absence of diaphragms, these fenestrations enable high fluid flux during kidney blood filtration with high water and small molecule solute permeability [8]. Non-diaphragmed and “discontinuous” ECs form the most permeable endothelial barriers due to the absence of a basement membrane and the presence of large sinusoidal gaps [12]. This unique structural feature allows for “discontinuous” endothelia with high permeability to water, small molecules, and macromolecules [13]. Examples of “discontinuous” ECs include liver sinusoidal ECs (LSECs) and bone marrow ECs.

### 2.2. Development

During vasculogenesis and angiogenesis, ECs uptake VEGF from their local microenvironment, which induces vascular network formation [14]. ECs can become post-mitotic as the cells constituting vascular networks mature; this is accompanied by downregulation of VEGF secretion by the surrounding epithelium [15]. However, in healthy tissues with fenestrated ECs (e.g., GECs, choroid plexus ECs), it has been found that the adjacent epithelium secretes VEGF persistently throughout maturation, coinciding with the expression of EC VEGF receptors (VEGFR)-1 and -2 [15,16]. As a result, it is widely believed that VEGF is the key regulator of endothelial fenestration development and maintenance. This has been experimentally shown through in vivo studies that showed that topical and intradermal administration of VEGF increased the permeability of postcapillary venules, muscular venules, and capillaries in mouse and rat models. VEGF (200 ng/mL) administration also led to the formation of endothelial fenestrations in the vascular beds of the cremaster muscle and skin, which normally are not fenestrated [17].

The role of VEGF signaling in endothelial fenestration has also been associated with PLVAP expression [18,19]. PLVAP creates a diaphragm (PLVAP+ diaphragm, which will be called “diaphragms” in the rest of this article) with a bicycle wheel-like frame that restricts permeability in most EC fenestrations (Figure 1B), except sinusoidal ECs and GECs. Although GECs form diaphragmed fenestrations during early kidney development before the glomerular basement membrane is assembled, such diaphragms are transient and are not present in mature GECs [20]. However, a recent study suggests that diaphragms may be re-expressed by GECs during cell damage and repair [6].

In addition to PLVAP expression, Rac was found to be involved in fenestration formation through VEGF signaling. A study by Eriksson et al. using mouse corneal angiogenesis models showed that VEGF-induced endothelial fenestration is dependent on phosphatidylinositol-3-OH kinase (PI3K)-Rac activation and an increase in phosphorylation of phospholipase Cγ (PLCγ), protein kinase B (Akt), and endothelial nitric oxide synthase (eNOS) [21] (Figure 1B). Given the regulatory role of Rac in cytoskeleton actin polymerization and organization [22,23], Nakakura et al. investigated the role of the actin cytoskeleton in endothelial fenestration formation using primary rat pituitary anterior lobe ECs. The authors found that inhibiting polymerization or promoting depolymerization of the actin cytoskeleton altered fenestration size, number, and arrangement [24]. Much like the plasma membrane, the actin cytoskeleton structure plays a critical role in endothelial fenestration formation (Figure 1B).

Despite this evidence regarding the role of the actin cytoskeleton in endothelial fenestration formation, the exact structural mechanism involved remains unclear. Satchell et al. suggested that fenestrations could originate from preexisting structures such as caveolae (expressing PLVAP) or vesiculo-vacuolar organelles, which fuse together with the apical and basal EC membrane to form opening pores through the cell cytoplasm [6] (Figure 1C). Alternatively, fenestrations could form de novo when the apical and basal cell membranes fuse in fenestration-forming centers without the presence of preexisting caveolae (Figure 1C). This de novo mechanism is believed to be involved in the development of non-diaphragmed fenestrations, such as LSECs and GECs [6,25].

Endothelial fenestrations play an important role in maintaining homeostasis in various organs and tissues, and their abnormal development has been indicated in the tumor microenvironment. In solid tumors, abnormal fenestrations can lead to increased vascular permeability known as the enhanced permeability and retention (EPR) effect, a pathophysiological phenomenon characterized by abnormal accumulation of macromolecules (>40 kDa) in the tumor vascular microenvironment [26]. The extravasation mechanism of such macromolecules is mediated by gaps between tumor ECs and their fenestrations [26,27]. Prior knowledge that tumors secrete VEGF to attract the surrounding vasculatures to invade and provide nutrients to the tumor microenvironment led to the identification of endothelial fenestrations in tumors by Roberts et al. in 1997 [28]. The authors found that supplementing tumors with VEGFA (a subtype of VEGF that is believed to be the major inducer of EC fenestration [29]) led to fenestrated endothelia in the tumors, whereas the surrounding vasculature in the skin and muscles was not fenestrated [28]. Unlike other studies showing fenestration formation in typically non-fenestrated ECs through VEGF signaling, this study highlights a tumor vasculature-specific response to VEGF-induced fenestration formation. A follow-up study by Grunstein et al. demonstrated the indispensable role of VEGF for the induction of endothelial fenestrations in tumorigenesis, where loss of VEGF expression resulted in decreased tumor vascular density, fenestration, and permeability, leading to tumor cell apoptosis [30]. Given the highly fenestrated nature of the tumor vasculature network and the resulting EPR effect, fenestrations have been widely investigated as a potential pathway to enhance targeted drug delivery to the tumor microenvironment [31]. The applications and challenges of the EPR effect in tumor drug delivery have been reviewed previously [26,32]. Additionally, considering the role of VEGF in the regulation of these aberrant fenestrations in tumors, VEGF inhibition has also been explored as a therapeutic target for cancer, resulting in various FDA-approved drugs, such as Axitinib (Inlyta^®^), Bevacizumab (Avastin^®^), and Cabozantinib (Cometriq^®^) [33].

### 2.3. Fenestrations in Different Organs

Researchers continue to elucidate the specialized characteristics of endothelial fenestration in organs critical for human development, maintenance, and susceptibility to disease. Efforts to better understand common and distinct fenestration structure and function have helped unveil mechanisms of vascular disease and guided the development of targeted therapies to inhibit degeneration or promote regeneration in fenestrated EC-populated organs/tissues, such as liver sinusoids [34] and kidney glomerulus [35]. In the next section, endothelial fenestrations in various organ/tissue types are discussed and summarized in Table 1.

#### 2.3.1. Brain

In the brain, molecular transport between the central nervous system (CNS) and the circulatory system is tightly regulated by the BBB, which is composed of a compact layer of non-fenestrated ECs, pericytes, astrocytes, and microglia (Figure 1D) [36]. Despite this tight barrier at the blood–CNS interface, the tissue constituting the blood–cerebrospinal fluid interface—known as the choroid plexus—contains a layer of highly fenestrated ECs (Figure 1D). Choroid plexus ECs line the luminal surface of the choroid plexus capillaries and are characterized by diaphragmed fenestrations [36]. The major function of these fenestrations is to provide rapid delivery of water to aid in cerebrospinal fluid production by the epithelial cells [37]. Additionally, given the high permeability of the choroid plexus capillaries compared to the BBB, the endothelial fenestrations of the choroid plexus provide a route for T cells to enter the cerebrospinal fluid. Active infiltration of lymphocytes into the cerebrospinal fluid through the choroid plexus has been demonstrated by Carrithers et al. [38], highlighting the role of choroid plexus EC fenestrations in supporting immune regulation in the CNS. Indeed, circulating lymphocytes in the choroid plexus capillaries can transmigrate across the fenestrated endothelia to enter the stroma of the choroid plexus, where macrophages and dendritic cells are located to present antigens to the lymphocytes. These lymphocytes can then migrate across the epithelium to enter the ventricles and mediate immune response regulation in the CNS (Table 1). Despite the significant roles of fenestrated choroid plexus ECs, their developmental process is poorly understood. Therefore, Parab et al. utilized a zebrafish model to elucidate choroid plexus EC development, and they found that a combination of VEGFAb, VEGFC, and VEGFD is crucial for zebrafish choroid plexus EC development, and Ccbe1—a proteolytic protein—was found to be involved in the activation of VEGFC [39]. However, the role of Ccbe1 in human choroid plexus EC development is yet to be validated.

#### 2.3.2. Gastrointestinal Tract

In the gastrointestinal tract, after nutrients are transported across the epithelium into the lamina propria during digestion, the nutrients are further transported into the underlying vasculature across diaphragmed endothelial fenestrations. In fact, blockage of VEGF signaling to reduce EC fenestration has been shown to reduce glucose adsorption [40].

Interestingly, unlike other endothelial fenestrations that exist on the entire luminal surface of the vasculatures, ECs in the small intestine villi vasculatures were found to have their fenestrations localized only on the exterior, epithelial-facing side, but the side facing the stroma is non-fenestrated (Figure 1D) [41]. A recent study performed by Bernier-Latmani et al. examined VEGF’s regulation mechanism in the small intestinal blood vessel microenvironment [41]. The authors found that a type of villus tip epithelial cells (perivascular *LGR5*+ villus tip telocytes) express protease ADAMTS18, which sequesters VEGFA by restricting fibronectin accumulation. This process maintains appropriate VEGFA levels in the intestinal villus blood vessel microenvironment to ensure sufficient endothelial fenestrations while maintaining villus tip structural integrity without excess fenestrations.

The gastrointestinal tract is continuously challenged by the presence of various antigens originating from commensal bacteria, pathogens, and food [42]. The mucus, intestinal epithelium, and intestinal endothelium function together to modulate antigen presentation to the immune system. The intestinal endothelial fenestrations were shown to participate in immune response modulations, as the presence of fenestrations allows for intercellular communication between intestinal epithelial cells and immune cells in the bloodstream (Table 1) [43]. Additionally, the intestinal endothelial fenestrations can be dynamically regulated by bacterial infection, as upregulation of PLVAP was observed after *Salmonella* infection in both mice and humans [44]. Such an increase in PLVAP expression leads to increased intestinal EC fenestration and vascular permeability, resulting in systemic dissemination of the bacteria to other organs, including the liver and spleen.

#### 2.3.3. Liver

LSECs line the walls of liver sinusoidal capillary channels. The sinusoidal capillary receives blood from the hepatic artery and the portal veins, to which oxygenated blood is provided by the hepatic artery, and nutrients and antigens are provided by the portal veins from the gastrointestinal system (Figure 1B). In the sinusoidal capillary system, blood from both the hepatic artery and the portal veins mixes and gets filtered by surrounding cells in the microenvironment. Subsequently, the mixed and filtered blood flushes into the hepatic vein to rejoin systemic blood circulation [45]. In the liver, the highly fenestrated and discontinuous LSECs play a crucial role in the maintenance of hepatic homeostasis and regulation of the hepatic immune response (Table 1) [5].

LSEC fenestrations were first observed by Wisse et al. in 1970 using transmission electron microscopy—clusters of LSEC fenestrations were found to form sieve plates in rats [46]. Initially, the diameter of rodent LSEC fenestrations was thought to range from 100 nm to 200 nm [5]. However, a more recent study by Szafranska et al. demonstrated that the tissue fixation approach and electron microscopy systems can affect LSEC fenestration size [47]. In this study, the authors found that dry-fixed samples for electron microscopy resulted in a 30–40% increase in fenestration size readout compared to wet-fixed samples (~173 ± 58 nm). As such, the authors proposed that future studies employ multimodal assessments including atomic force microscopy, scanning electron microscopy, or stimulated emission depletion microscopy for a more accurate quantification of fenestrations.

While a tight EC layer (with positive expression of CD31 and CD144) lines the luminal surface of the vasculature on top of a basement membrane layer that serves as structural support, the liver endothelium exhibits different morphological features. LSECs possess non-diaphragmed and discontinuous fenestrations, and they lack an intact basement membrane. Additionally, LSECs do not consistently express CD31 and CD144 [48]—instead, they express LYVE-1, CD32, CD36, CD14, and CD54, the co-expression of which is essential for identifying LSECs [48]. These features create high permeability to water, small solutes, and macromolecules, such as albumin and fibrinogen [49]. The highly fenestrated structure also allows for bidirectional passive transport of various molecules, including lipoproteins, drugs, and metabolites, between the blood and the liver parenchyma. Additionally, LSEC fenestrations allow circulating T cells to interact with underlying hepatocytes by extending protrusions across the LSEC fenestrations [50]. Moreover, monocytes have been shown to move bidirectionally across LSEC fenestrations; a migration process involved in monocyte fate decision that leads to the formation of macrophage-like cells in the local tissue and release of pre-dendritic cells to the circulation [51,52]. Reverse transmigration into luminal space tends to generate pro-inflammatory CD16+ monocytes, indicating the important role of LSEC fenestration in immune regulation and inflammation.

Liver injury or aging can cause LSECs to lose these specialized features and transition toward a capillary phenotype in a process called capillarization (or pseudocapillarization in aging livers) [53]. This process involves a decrease in fenestration size and number in addition to a thickening of the endothelium and deposition of collagen in the space of Disse (or perisinusoidal), restricting molecular transport between the blood and hepatocytes [54]. Such a loss of LSEC fenestrations can hinder insulin and lipoprotein transport, leading to hepatic insulin resistance, reduced clearance of insulin, diabetes, dyslipidemia, and atherosclerosis [55]. It has been further suggested that pseudocapilliarization may lead to a restriction of oxygen diffusion to hepatocytes, leading to intracellular hypoxia [56,57]. Additionally, a reduction in LSEC fenestrations can lead to the limited passage of drugs for processing and metabolism, therefore resulting in the accumulation of drugs in the liver and elevated drug toxicity [54]. For instance, Hilmer et al. demonstrated that passage of the chemotherapeutic liposomal doxorubicin through the sinusoidal endothelium is restricted in aging rat livers as compared with younger rats, likely due to pseudocapillarization [58,59]. This reduces the opportunity for hepatic extraction of the drug and may account for the altered pharmacokinetics of liposomal doxorubicin as compared with its unencapsulated counterpart doxorubicin. The impact of fenestration loss on the metabolism of large molecule therapeutics and liposomal encapsulated drugs raises important concerns for patients with liver disease and the elderly. Reversing pseudocapillarization is a growing area of interest, and pharmacological candidates such as nicotinamide mononucleotide, sildenafil, and 7-ketocholesterol have shown promise in increasing fenestration porosity and frequency [34].

#### 2.3.4. Bone

Bone marrow is vital to human health as it is the critical site for hematopoiesis, or the production and maturation of new blood cells. The release of mature blood cells into peripheral circulation is facilitated by the unique bone vasculature. Vessels of the bone marrow have distinct structures and function, depending on the region in which they reside. Distinct from type-H ECs, type-L ECs have the lowest expression level of CD31 and endomucin cell surface markers and are located in the diaphysis, or main shaft [60]. Importantly, type-L ECs have sinusoidal pores. The sinusoids are well known for facilitating osteogenesis and hematopoiesis through the trafficking of cells, delivery of oxygen and nutrients, and promotion of angiocrine factors to mediate regeneration (Figure 1D). While sinusoidal ECs of other organs lack adventitia, bone marrow ECs are uniquely supported by stromal cell-derived factor-1-expressing adventitial cells and clusters of myeloid cells that mature into specific circulating cell types (e.g., macrophages, neutrophils, erythrocytes, etc.). The basement membrane of bone marrow ECs is discontinuous which allows for the egress of cells and penetration by other molecules (Table 1). Additionally, the structural characteristics of sinusoidal vessels give rise to distinct fluid flow dynamics compared to arterial vessels, as determined by Bixel et al. [61].

At this point, diseases that are directly caused by structural abnormalities of the bone marrow sinusoidal ECs have not been described, yet researchers are starting to uncover the important effect of common morbidities on the vasculature that may lead to a better understanding of the molecular interplay between bone marrow homeostasis and other organs to provide more accurate therapies and reduce off-target effects. For example, β-catenin dysregulation has been shown to occur in several diseases, including cancer (e.g., colorectal carcinoma, liver carcinoma, leukemia, etc.) and neurological disorders (e.g., Alzheimer’s disease, Parkinson’s disease, multiple sclerosis, etc.) [61]. Heil et al. found that constitutively active β-catenin blocked terminal erythroid differentiation, increased deposition of extracellular matrix (ECM) molecules in the perisinusoidal basement membrane, and activated hematopoiesis outside of the bone marrow vasculature (in the liver and spleen), resulting in lethal anemia [62]. Similarly, infection can cause hematopoiesis outside of the bone marrow vasculature, which can cause bleeding and polyps in the gastrointestinal tract, but this mechanism is still poorly understood. In addition to dysregulation of hematopoiesis, bone marrow sinusoids are affected by other major organs such as the heart. Rohde et al. showed that hypertension, atherosclerosis, and myocardial infarction of the heart increase bone marrow EC dysfunction partially via increased arteriole and sinusoid vessel density that resulted in leakage, fibrosis, emergency angiogenesis, and, ultimately, overproduction of inflammatory myeloid cells. The results suggest the signaling pathways involved VEGF since the inhibition of VEGFR-2 limited the increase in bone marrow arterioles and sinusoid density in the femurs of mice following myocardial infarction [63]. In summary, while we understand the dire consequences of damage to the bone marrow stem cell niche, there is a need to discover a direct tie between bone marrow sinusoidal EC dysfunction and disease pathogenesis. However, we are starting to uncover the mechanisms of bone marrow EC dysfunction that stems from common comorbidities such as cancer, infection, and cardiovascular disease that can contribute to the promotion of more precise therapeutic strategies with fewer off-target effects.

#### 2.3.5. Retina

The blood–retina barrier (BRB) regulates oxygen and nutrient supply to photoreceptors and the glial cells of the retina, which maintains visual perception. Specifically, the inner BRB is composed of pericytes, glia, and retinal ECs, whereas the outer BRB consists of retinal pigment epithelial (RPE) cells, Bruch’s membrane (a 5-layer composition of ECM protein that acts as a molecular sieve), and choriocapillaris (capillary network) [64]. Compared to other capillaries, the choriocapillaris have a larger luminal surface to optimize the exchange of nutrients and removal of waste (Figure 1D, Table 1) [65]. This essential function is mediated through specialized components of the cellular membrane, including caveolae, vesiculo-vacuolar organelles, coated pits, and fenestrations (on the retinal side) possessing diaphragms. Recent reports are helping to illuminate the relationship between PLVAP and retinal integrity. Kim et al. found that VEGFA maintains the endothelial PLVAP ultrastructure, while PLVAP loss results in retinal degeneration [66]. In a mouse model of sodium iodate (NaIO_3_)-mediated photoreceptor degeneration, VEGFA gene expression decreased after 1 day of NaIO_3_ injection, and this was quickly followed by the loss of fenestrated choriocapillaris and complete disappearance of the PLVAP immunostaining signal (2- and 7-days post-injection, respectively) [67]. However, the non-diaphragmed fenestrations of the GEC were unaffected. Independently, Ida et al. described the loss of the choriocapillaris fenestrations as seen in an experimental model of age-related macular degeneration (AMD) [68]. The authors described the effects of D-galactose treatment on the outer BRB of an advanced aging mouse model. Their results suggest that the choroidal ECs in mice are altered more than the RPE and that the increased presence of advanced glycation end products, or AGEs, could play a pivotal role in disease development due to their ability to mediate pro-inflammatory signals. These findings have implications for several age-related diseases, such as atherosclerosis and Alzheimer’s disease. It is critical to uncover and therapeutically target pathways of chronic inflammation that may be structurally and molecularly facilitated by the choriocapillaris’ unique fenestrations and other retinal ECs.

#### 2.3.6. Kidneys

In the kidneys, blood is initially filtered across the glomerulus, and the ultrafiltrate passes through the renal tubules, where water, salts, and nutrients are reabsorbed back into the blood while excess water and metabolic wastes are eventually excreted as urine. Due to the extensive involvement of bulk water movement in glomerular filtration and tubular reabsorption, ECs in both the glomerulus and tubules are highly fenestrated. However, the major distinction between GECs and peritubular ECs is that GEC fenestrations are non-diaphragmed, while peritubular endothelial fenestrations are diaphragmed.

In the glomerulus, GECs line the luminal surface of the glomerular capillaries and form an interface (separated by a thin layer of basement membrane) with highly specialized epithelial cells known as podocytes (Figure 1D and Figure 2). Together, the glomerular basement membrane, podocytes, and GECs constitute the glomerular filtration barrier. In early embryonic kidneys, GECs do not exhibit fenestrations. As the embryonic kidney matures, VEGF signaling from the cell’s local microenvironment induces the formation of diaphragmed GEC fenestrations, as observed in embryonic rat GECs [25]. However, mature GECs are characterized by non-diaphragmed fenestrations. The loss of fenestration diaphragms in mature kidneys can be attributed to the involvement of endothelial A disintegrin and metalloprotease domain 10 (ADAM10), a notch signaling regulator, as ADAM10-deficient mice demonstrate persistent GEC fenestration diaphragms compared to control mice [69]. Apart from VEGF and ADAM10, Eps15 homology domain-containing protein (EHD3) has been shown to regulate GEC fenestration formation. In the kidneys, EHD3 is specifically expressed in GEC fenestrations; knockdown of EHD3 expression leads to loss of GEC fenestrations and knockout of EHD3 leads to absence of GEC fenestrations in mouse models [70,71]. As endothelial fenestration formation is closely associated with actin cytoskeleton arrangement, Ezrin, radixin, and moesin (ERM)—proteins interacting with transmembrane proteins and the cytoskeleton—are suggested to be involved in GEC fenestration formation through chloride intracellular channel protein (CLIC)CLIC5A- and CLIC4-induced phosphorylation and downstream actin remodeling. Knockout of CLIC5A and CLIC4 in a mouse model reduced ERM phosphorylation, accompanied by a loss of GEC fenestrations [72]. However, most existing studies have been focused on small animal models, and validating these targets in human biopsy samples would provide valuable insights. Apart from the non-diaphragmed feature, GEC fenestrations also exhibited relatively large sizes ranging from 60 to 80 nm in diameter [73]. As a result, it was historically believed that GECs did not contribute to the selective filtration function of the glomerular filtration barrier until the discovery of glycocalyx on the GEC luminal surface in 2008 [74]. Glycocalyx is a negatively charged proteoglycan that anchors to the cell surface and forms a hydrated mesh-like network capable of adsorbing glycosaminoglycans in the plasma. Indeed, it is well documented that fenestrations are coated with glycocalyx [73]. Now, it is recognized that all three components of the glomerular filtration barrier (podocytes, glomerular basement membrane (GBM), and GECs) form an integrated functional unit to regulate size-selective filtration. While podocytes form interdigitated slit diaphragms to prevent the passage of large molecules (e.g., albumin) and the glomerular basement membrane exhibits a net negative charge to expel negatively charged proteins [75], the fenestrations in GECs contribute to fluid flux and the passage of small solutes, and the glycocalyx coating the fenestrations also helps to restrain the passage of large molecules across the filtration barrier (Table 1). Defects in any of the three GBM components can lead to disruption of the overall permeability, resulting in proteinuria, a pathological condition characterized by the loss of large proteins in the urine.

**Table 1 ijms-25-09107-t001:** Summary of fenestrated ECs in various organs/tissues and their corresponding specificity in terms of structural features and specific functions.

Organ/Tissue Type	Diaphragmed?	Discontinued?	Basement Membrane?	Primary Functions	References
Choroid plexus	Yes	No	Yes	Deliver water for cerebrospinal fluid production; Facilitate lymphocyte infiltration.	[36,37,38]
Small intestine	Yes	No	Yes	Nutrient transport; Facilitate immune cell-intestinal epithelial cell crosstalk.	[41,43]
Liver sinusoids	No	Yes	No	Molecular transport across live sinusoids to facilitate liver metabolism and homeostasis; Promote T cell interaction with hepatocytes; Trans-endothelium migration of monocytes.	[45,49,50]
Bone	No	Yes	Yes	Facilitate osteogenesis and hematopoiesis through trafficking of cells, delivery of oxygen and nutrients, and angiocrine factors to mediate regeneration	[60,61]
Retina	Yes	No	Yes	Regulates oxygen and nutrient supply to photoreceptors and glial cells of the retina, which maintains visual perception	[64,65,66]
Kidney glomerulus	No	No	Yes	Facilitate water and small solute transport across the glomerular filtration barrier.	[73,75]
Kidney peritubular capillaries	Yes	No	Yes	Regulation of water, protein, and small molecule transport in the interstitium.	[76,77]

The human kidney peritubular microvascular ECs (HKMECs) line the luminal side of the peritubular capillaries that provide oxygen and nutrients to tubular and interstitial cells in the kidney cortex (Figure 1B) [76]. HKMECs also possess fenestrations that are 60–80 nm in diameter, but unlike GEC fenestrations, they contain diaphragms that regulate the transportation of water, proteins, and other molecules in the interstitium (Table 1) [77]. Damage to the peritubular capillaries leads to the progression of kidney disease (e.g., diabetic nephropathy [78] and hypertensive nephrosclerosis [77]) and loss of kidney function [77]. Injury of HKMECs has been implicated in tissue ischemia, tubular dysfunction, inflammation, and fibrosis [79]. However, most of the existing studies focus on the rarefaction of peritubular capillaries [80,81,82], and little is known about the exact roles of peritubular endothelial fenestrations.

In aging kidneys, function naturally declines, with worsening cases in adults over 70 years old [83]. However, the cause of this age-related decline in glomerular filtration rate (GFR) and its potential connection to endothelial fenestrations is unknown [8]. Considering the established precedence of reduced liver function caused by fenestration loss seen in pseudocapillarization (described in Section 2.3.3), it is possible that loss of endothelial fenestrations with aging may result in a similar reduction in kidney function. This is an untapped area of research that should be further explored.

### 2.4. Implications for Kidney Disease

Chronic kidney disease (CKD) is a global health problem that affects around 13% of the world population and 15% (1 in 7) of the US population [84,85]. CKD is one of the leading causes of death globally [86]. The high prevalence of the disease is compounded by the lack of early biomarkers, effective therapies to stop its progression into ESKD, and comorbidities [87]. CKD is characterized by progressive structural and functional changes to the kidneys due to various causes including viral infections, drug toxicity, genetic mutation, and lifestyle [88]. Progression of CKD is partly associated with the gradual loss of GEC fenestrations [89]. The reappearance of PLVAP+ diaphragms due to disturbed intercellular crosstalk between GECs and podocytes, especially VEGFA signaling, has also been implicated in progressive CKD [90]. The loss of endothelial fenestrations can lead to a decline of glomerular filtration function and a buildup of uremic toxins in the body. Pathological changes in GEC fenestrations in specific types of kidney diseases are discussed below.

#### 2.4.1. Diabetic Nephropathy

Diabetic nephropathy (DN) is a common complication of diabetes, with up to 40% of diabetic patients developing the condition. DN leads to ESKD, which requires dialysis or kidney transplantation, and the mortality rate of diabetic patients with DN is 30 times higher than that of those without it [91,92]. DN patients usually present with hyperfiltration and albuminuria in the early phases of the disease, which subsequently develop into progressive loss of glomerular filtration function [93]. Efforts are currently being made to understand the disease mechanisms of DN to help improve disease management and slow its progression to ESKD. For example, although angiotensin-converting enzyme (ACE) inhibitors have been considered a standard of care for hypertension in diabetic patients, with no known effects on endothelial fenestrations [94], sodium-glucose cotransporter-2 (SGLT2) inhibitors have recently been identified as a promising therapy to slow DN progression [95]. Locatelli et al. showed that a type of SGLT2 inhibitor (Empagliflozin) improves GEC ultrastructure by demonstrating downregulation of endothelial fenestration diaphragms and expression of PLVAP and caveolin-1 [96]. However, therapies to cure DN are still lacking, and a better understanding of DN pathogenesis combined with early diagnostics tools could aid the development of more efficient therapies.

Previous studies have shown that type 1 diabetes patients with multiple clinical presentations (normoalbuminuric, microalbuminuric, proteinuric) demonstrate a significant reduction in the glomerular capillary luminal surface covered by fenestrated endothelia compared to healthy individuals (Figure 2) [97]. A subsequent study by Weil et al. investigated changes in endothelial fenestrations in type 2 diabetes patients [98]. In this study, the authors found that the mean percentage of endothelial fenestration in the kidney glomerulus was significantly lower in diabetic patients (macroalbuminuria, 19.3%; normoalbuminuria 27.4%; microalbuminuria, 27.2%) compared to healthy individuals (43.5%). However, the authors were cautious not to draw a conclusion relating the reduction in endothelial fenestration to aberrant glomerular filtration.

A recent study by Finch et al. bridged this missing link by examining ultrastructural change in endothelial fenestrations in type 2 diabetic mice and type 2 diabetic human kidney biopsy samples and comparing their glomerular filtration functions [35]. The authors found that glomerular ECs in diabetic samples exhibited diaphragm formation in their fenestrations, which increased fluid filtration resistance and led to a decline in glomerular filtration function (Figure 2). Additionally, in diabetic mice models, there was a significant reduction in fenestration density accompanied by an increase in fenestration width. In patient samples, there were similar changes in glomerular endothelial fenestrations in addition to a significant reduction in fenestration surface area. An in-depth analysis of the diseased fenestration phenotype revealed a downregulation in EHD3 in patients with diabetes, confirming the regulatory role of EHD3 in glomerular fenestration formation and maintenance as observed previously in mouse models [35].

Given the important role of endothelial fenestrations in DN, targeted therapies aiming to restore and regulate fenestrations could provide a promising avenue to better manage and treat this disease. For example, Onions et al. showed that overexpression of VEGFC in both type 1 and type 2 diabetes mice models was protective against the loss of endothelial fenestrations while also reducing albuminuria [99]. In this study, VEGFC acted as a milder VEGF signaling mediator, as VEGFA was shown to be upregulated in DN, and this overexpression was thought to contribute to glomerular EC dysfunction (Figure 2). This study highlighted the importance of understanding GEC fenestrations, guiding the development of potential fenestration-targeting therapies for kidney diseases.

#### 2.4.2. Focal Segmental Glomerulosclerosis

Focal segmental glomerulosclerosis (FSGS) describes kidney diseases that are characterized by both primary podocyte injury and secondary lesions, ultimately progressing toward podocyte loss and tissue scarring [100]. A recent study performed by Tampe et al. characterized glomerular endothelial fenestrations in FSGS patient biopsy samples [101]. In this study, PLVAP, which is normally absent in GECs, was expressed in one-third of all the FSGS biopsy samples. Additionally, notable changes in GEC fenestrations were shown, including the loss of pores and formation of diaphragms, similar to those observed in DN (Figure 2). The authors suggested that the formation of diaphragmed fenestrations in diseased kidneys could be a compensatory mechanism to help prevent the loss of proteins into the glomerular filtrate. Additionally, another study by Morita et al. demonstrated that the changes in GEC fenestrations are lesion type-dependent in FSGS patients, and loss of fenestration and EC swelling were observed only when the segmental lesion involves the tubular pole (areas near the proximal tubule) of the glomerulus [102]. In experimental FSGS models, the progression of the disease was characterized by the initial loss of endothelial fenestrations followed by loss of glycocalyx and nitric oxide synthase activity; these factors further contributed to podocyte injury and loss of the glomerular filtration barrier integrity [103]. Nonetheless, the exact role of pathological changes in endothelial fenestrations in FSGS requires further studies.

#### 2.4.3. Alport Syndrome

In addition to DN and FSGS, endothelial fenestration loss has been indicated in Alport syndrome (AS), a rare genetic disorder associated with kidney disease. AS is caused by mutations in the *COL4A3*, *COL4A4*, and *COL4A5* genes that encode the α3, α4, and α5 chains of type IV collagen, which is a key component of the glomerular basement membrane [104]. AS patient biopsy samples exhibit splitting and enlargement of the glomerular basement membrane as well as podocyte effacement [104]. The role of GEC fenestrations in AS was demonstrated in AS mouse models by Sedrakyan et al., who showed that GECs had significantly increased fenestration size and expression of PLVAP [105]. Additionally, morphological changes in GEC fenestrations can be observed in patient biopsy samples as AS progresses [106,107,108]. However, the specific roles of these changes in AS progression are poorly understood. Elucidating the roles of GEC fenestrations in AS could shed light on disease progression, aid in the development of more efficient targeted therapies, and improve patient outcomes.

#### 2.4.4. Preeclampsia

Preeclampsia is a pregnancy-related disorder characterized by sudden onset and persistent high blood pressure with or without proteinuria, as well as thrombocytopenia, high liver transaminase levels, pulmonary edema, and cerebral or visual issues. Preeclampsia accounts for 2–8% of pregnancy-related complications, including 50,000 maternal deaths and over 500,000 fetal deaths worldwide each year [109]. In the kidney, glomerular endotheliosis from preeclampsia is a microangiopathy characterized by cellular hypertrophy, occlusion of the capillary lumens, and severe reduction of fenestration density and area and basement membrane coverage [8,110]. Some case reports and cohort studies have suggested that live kidney donors are at an increased risk of developing preeclampsia (1–3% pre-donation to 4–5% post-donation) [111]. A recent study in mice explored potential causes for this link [102]. Uninephrectomized pregnant mice exhibited maternal preeclampsia-like symptoms, including enlarged ECs that lacked fenestrations, placental ischemia, upregulation of soluble fms-like tyrosine kinase 1 (sFLT1, a VEGF antagonist), and dysregulated l-kynurenine (part of the anti-hypertensive response pathway). However, reduction of sFLT1 levels via antagonistic inhibition with placental growth factor or supplementation of l-kynurenine successfully rescued abnormal placentation, improved endothelial lesions, and preserved fenestrations (as observed via electron microscopy) [112]. These findings demonstrate the importance of diagnostic testing and counseling to help address issues related to high-risk pregnancies. Regulation of EC fenestrations could play a significant role in curtailing the manifestation of preeclampsia. Understanding key fenestration and hemodynamic modulators (e.g., VEGF, PLVAP, Angiotensin) could also provide additional molecular targets for addressing pregnancy-related complications in the future.

## 3. Future Perspectives

### 3.1. Leveraging Specialized EC Phenotypes for In Vitro Kidney Disease Modeling

While the current understanding of ECs is likely to improve in the future, it is recognized that the vascular endothelia is a key player in the regulation of homeostasis, as well as disease initiation and progression; therefore, high-fidelity in vitro models of the vascular endothelia are critical to improving understanding of organ function in health and disease [113]. The recent development of advanced microphysiological systems, such as spheroids, organoids, and organs-on-chips, has greatly improved our capability to model kidney function in vitro [114,115]. Vascularized kidney organoids show increased maturation and cell type complexity [116]. However, the heterogenicity and tissue specificity of ECs is often overlooked in most in vitro models, making it difficult to establish platforms that accurately represent constituents of different organs and their respective disease phenotypes. The highly specialized fenestrated ECs are one of the hallmarks of renal capillaries, and changes in EC fenestrations have been implicated in multiple renal diseases. Efforts to establish in vitro models that can accurately recapitulate human physiological responses must consider organ-specific EC phenotypes to fully understand the roles EC fenestrations play in disease onset and progression and enhance translational research studies in the kidney and across organs [13].

For instance, in vitro modeling of fenestrations by Finch et al. was vital to elucidating the cause of GEC fenestration loss and subsequent reduction in filtration function seen in patients with DN [35]. As described in Section 2.4.1, the authors utilized in vivo models to reveal a potential regulatory role of EHD3 in glomerular fenestration formation and function, hypothesizing a disease mechanism for DN in which EHD3 protein (an endosomal transport protein expressed in GEC fenestrations) expression is downregulated, leading to dysregulation of endothelial fenestrations and a decline in glomerular filtration function [35]. To prove causation between the loss of EHD3 and fenestration dysregulation, the authors turned to an in vitro model, knocking down EHD3 expression in b.End5 cells (a fenestration-forming cell line of mouse brain endothelioma cells). The control and knockdown cells were then supplemented with latrunculin A or VEGFA for the induction of fenestrations. In this in vitro model of EC fenestrations, the authors were able to show that reduced expression of EHD3 resulted in decreased fenestration formation in response to latrunculin A and the absence of fenestration formation in response to VEGFA.

While this result supported the authors’ proposed mechanism of DN, the capability to model GEC fenestrations in vitro also uniquely provided further insights into the previously unknown mechanistic role of EHD3 in fenestration formation [86]. The complete absence of fenestration formation in response to VEGFA supplementation compared to the reduction of fenestrations in response to latrunculin A supplementation supported the hypothesis that EHD3 regulates fenestrations via endocytic recycling of VEGFR2. Previous studies suggest that endocytic recycling protects VEGFR2 from plasma membrane cleavage, thereby maintaining VEGFR2 and VEGF signaling, which are essential to fenestration development and function [117,118]. Although the authors recognized that b.End5 cells are not an ideal model or an appropriate system for comparison to human GECs, these mechanistic insights set the stage for further investigations.

### 3.2. Challenges in Modeling Kidney EC Fenestrations In Vitro

Traditional in vitro models of endothelial fenestrations lack physiological relevance to the in vivo environment of the kidney glomerulus [8]. These models often include human primary GECs, which have been shown to exhibit fenestrations under appropriate culture conditions [119], or conditionally immortalized GECs, for which certain growth factors such as VEGF supplementation can induce fenestrations [8,120]. Additionally, fenestration-forming cell lines such as b.End5 have been used as an approximation for fenestration behavior in GECs [8,35]. However, confidence in the relevance of these cell lines to human GECs is limited considering the wide range of EC types and limited data on their functional and molecular characteristics compared to human GECs in vivo. For instance, the efficiency of VEGF receptor signaling activation can be species-dependent, with mouse VEGF being more efficient for signal transduction in mouse endothelial cells and human VEGF being more efficient for signal transduction in human endothelial cells. Therefore, the use of human VEGF in studies relying on mouse cells such as b.End5 may have different dose-dependent effects [121].

Importantly, single-monolayer cell culture models do not account for the dynamic interactions between GECs and podocytes. VEGF secreted by podocytes has been attributed to GEC fenestration induction in vivo and in vitro [6,122]. GEC fenestration maturation, in which GECs lose their fenestration diaphragms during glomerular development, has also been shown to occur concurrently with the maturation of podocyte foot processes [6].

Moreover, fenestrations are extremely sensitive to mechanical cues in the EC environment, which cannot be replicated in traditional cell culture plate models. In fact, shear stress has been shown to induce fenestrations in GECs and plays a key role in EC development [6,123]. Wall shear stress created by blood flow and transduced through the glycocalyx can regulate the transcriptional activity of eNOS and the production of NO, thereby impacting the VEGFA-eNOS/NO-endothelin 1 (ET-1) axis and regulating the expression of VEGFA [90]. During proper functioning of VEGFA-eNOS/NO-ET-1 signaling, secretion of VEGFA by podocytes induces activation of eNOS, positively regulating NO. Meanwhile, NO negatively regulates VEGFA, maintaining the delicate balance of VEGFA levels necessary for proper fenestration regulation. A recent study by Bevan et al. demonstrated the impact of laminar shear stress on GEC permeability in vitro [124]. In this study, Bevan et al. exposed human conditionally immortalized GECs to physiologically relevant levels of shear stress (10–20 dyn/cm^2^) for 24 h via an orbital shaker and measured the resulting transendothelial electrical resistance as an indicator of permeability. Exposure to laminar shear stress of 10 dyn/cm^2^ resulted in a significant and reversible increase in GEC permeability. The authors found this permeability increase to be dependent on NOS production through the PI3K/Akt pathway. It is important that models of GEC fenestrations account for shear stress, both for improving physiological relevance and to support future research on how diseases that affect shear stress, such as atherosclerosis, might impact fenestration size, frequency, and structure.

Shortcomings of modeling fenestrations in 2D culture environments have impacted the fidelity of traditional in vitro models of GECs. For example, fenestrations induced in single-monolayer cell culture models do not form organized sieve plates [8,125]. In addition, many cell lines that naturally contain fenestrations in vivo have been shown to express a lower density or even no fenestrations in a 2D in vitro culture, and more still lose their fenestrations over time [11]. While primary GECs cultured under basal conditions have been shown to contain pores that are believed to be fenestrations, the pores are distributed at a much lower density and have a different appearance than fenestrations in vivo. Human GECs in a 2D in vitro culture also behave differently than those in vivo. When exposed to the fenestration-altering cytokine tumor necrosis factor, which has been shown to reduce VEGF signaling, human GECs in culture develop an increased number of smaller fenestrations, while human GECs in vivo exposed to a similar condition show fewer fenestrations with a larger width [8,126,127]. These differences are evident in the phenotypic drift that often occurs when ECs are removed from their native in vivo environment. It also highlights limitations in the field regarding knowledge of all the molecular and biophysical signals necessary to accurately model fenestrated EC biology.

While a growing number of reports have explored the biology and physiology of GEC fenestrations, other EC types in the kidneys remain substantially understudied [79]. Another key fenestrated cell type in the kidneys is HKMEC. Primary HKMECs in monoculture have shown little fenestration formation, as evidenced by primarily perinuclear expression of PLVAP [128]. However, VEGF partially improves PLVAP expression in HKMECs cultured in monocultures, suggesting a regulatory mechanism similar to GECs. Still, the mechanism of fenestration formation in primary HKMECs in monocultures is not fully understood.

The study of fenestrations in kidney cells can no longer focus on GECs in 2D in vitro cultures. Future studies should take advantage of new technologies, such as microphysiological models, which are capable of more accurately modeling the kidney microenvironment, including intercellular crosstalk, mechanical cues, fluid dynamics, and basement membrane content and structure. Models of the kidney must also expand beyond the glomerular capillaries to include the peritubular capillaries and surrounding vasculature, such that we may observe the impact of EC specialization on fenestration structure and function.

### 3.3. Emerging Microphysiological Models of Kidney EC Fenestrations

Microphysiological models have the potential to improve our understanding of fenestrations in kidney ECs by recapitulating the interactions between different cell types, as well as the mechanical and biophysical cues experienced by cells and tissues, and the 3D microenvironment in the kidneys. In a recent study, Rederer et al. [115] co-cultured human primary GECs, immortalized human podocytes, and human primary mesangial cells in 3D agarose micro-wells, resulting in spheroid self-assembly. The GEC cells in these spheroids demonstrated fenestration-like structures as characterized by Scanning Electron Microscopy. The authors also demonstrated (via fluorescence colocalization of green fluorescent podocyte-derived VEGFA and red fluorescent tdTomato-Farnesyl GECs) the transport of podocyte-derived VEGFA to GECs. It is likely that the observed fenestrations were partly induced by this podocyte-derived VEGFA [6,122].

Pajoumshariati et al. similarly co-cultured human primary GECs, podocytes, and human primary mesangial cells [129]. In contrast to the work of Rederer et al., this group utilized human induced pluripotent stem cell (hiPSC)-derived podocytes and interfaced the different cell types in a glomerulus-on-a-chip. By adding a 3D hydrogel in the bottom channel, Pajoumshariati et al. increased the shear stress in the bottom channel (0.136 dyn/cm^2^) as compared with chips without a 3D hydrogel (0.017 dyn/cm^2^). This 8-fold increase brought the shear stress experienced by the GECs closer to that experienced by glomerular cells in vivo (0.7–1.2 dyn/cm^2^). This was expected to yield GECs that more closely resembled GECs in vivo, given the importance of shear stress in EC development (Figure 3). Pajoumshariati et al. [129]. claimed that their model would support fenestrations in GECs, as evidenced by the expression of EHD3. However, they did not perform high-resolution electron microscopy characterization of their model; therefore, it remains unknown whether fenestrations were actually formed or existent in their system.

Zhang et al. co-cultured primary HKMECs and primary human kidney proximal tubule epithelial cells (HPTECs) in an open microfluidic device. In this device, HKMECs and HPTECs were cultured in separate chambers divided by a polystyrene half wall. Additional culture media was then added to overflow the half wall and initiate paracrine signaling between the two chambers. Zhang et al. demonstrated that the co-culture of HKMECs with HPTECs in this device induced clear clusters of PLVAP expression indicative of robust fenestration formation [128]. However, as in the report by Pajoumshariati et al., Zhang et al. did not confirm the formation of fenestrations with electron microscopy [128,129].

These preliminary studies involving microfluidic systems require further validation with high-power electron microscopy techniques (such as Transmission Electron Microscopy and Scanning Electron Microscopy) to provide concrete evidence for the development of EC fenestrations in their in vitro models. High-power electron microscopy techniques, while the gold standard for validation of endothelial fenestrations, are a challenge for many studies of microphysiological models. One of the most common membrane materials used in organs-on-chips, polydimethylsiloxane, is often difficult to section with the microtome as required for high power Transmission Electron Microscopy [122,130]. Research into alternative sectioning techniques or membrane materials could greatly advance capabilities to study fenestrations in microphysiological models.

Microphysiological systems with fenestrated HKMECs have already supported improved disease modeling. For instance, a 3D microvascular network engineered in collagen gel in the presence of HKMECs and VEGF supplementation and cultured under gravity-driven fluid flow developed numerous fenestrations similar in width to HKMEC fenestrations in vivo [79]. This model was later utilized to demonstrate that cyclosporine A induces renal microvascular injury [131]. Cyclosporine A is often used in organ transplants as an immunosuppressant. However, exposure to cyclosporine A has been implicated in renal vascular injury, thrombotic microangiopathy, and striped interstitial fibrosis. Work by Nagao et al. revealed that cyclosporine A impairs VEGF signaling in HKMECs, inhibiting activation and nuclear translocation of NFAT1 and, ultimately, leading to retractions between adjacent HKMECs and disruption of fenestration structure [131]. Cyclosporine A treatment disrupted the formation of fenestration diaphragms, leading to closure or full opening of the membrane and shedding of PLVAP. This loss of fenestrations, as shown by this model, is hypothesized to contribute to impaired kidney function. Ligresti et al.’s model of HKMECs in a 3D environment under perfusion was vital to the work by Nagao et al., who demonstrated shear-dependent dissociation of cyclosporine A-treated HKMECs [79,131]. Still, the role of endothelial fenestrations in other kidney diseases such as DN and FSGS has yet to be studied in microphysiological models. Systematic studies of kidney diseases in microphysiological models could improve the current understanding of how the interaction between endothelial fenestrations and nearby epithelial cells in addition to the effect of mechanical cues and shear stress is altered in diseased states.

### 3.4. Towards hiPSC-Derived Models of Fenestrated Kidney ECs

Although preliminary efforts have been made in establishing microphysiological systems to model kidney EC fenestrations using primary cells, the ability to develop in vitro methods for inducing hiPSCs into fenestrated GECs will pave the way for patient-specific models of kidney function (Figure 3) [132]. Unlike primary cells and other established cell lines, hiPSCs could serve as an unlimited and on-demand supply of cells for modeling human development, physiology, and pathophysiology given their ability to self-renew indefinitely and differentiate into almost any cell type when given appropriate cues [133]. However, methods to differentiate hiPSCs into organ-specific ECs are yet to be fully described.

Initial attempts to induce fenestrations in hiPSC-derived ECs have largely relied on rodent kidneys. One such method, developed by Ciampi et al., involved repopulation of a decellularized rat kidney scaffold with hiPSC-derived ECs [134]. This approach appeared to induce fenestration development in the glomerular capillaries but not in vascular capillaries, indicative of site-specific endothelial specialization. Renal subcapsular transplantation of hiPSC-derived kidney organoids has also been shown to induce fenestrations in hiPSC-derived ECs [135]. However, these methods cannot be classified as in vitro induction of fenestrations, as they rely on implantation in vivo or in ex vivo tissues. Additionally, the development of rodent and human kidneys varies widely, including differences such as the length of nephrogenesis and final nephron count, which some attribute to differences in *Six1* and *Six2* regulation [136]. This difference in kidney structure and molecular signaling pathways between species decreases the physiological relevance and translational power of irrelevant models.

Meijer et al. recently disrupted this traditional reliance on rodent models by creating the first published method by which fenestrations could be reliably induced in hiPSC-derived ECs entirely in vitro [133]. The authors demonstrated that ECs extracted from hiPSC-induced vascular organoids formed fenestration in tissue culture plates through supplementation with VEGFA or phorbol myristate acetate (PMA) (Figure 3). PMA has been shown to activate protein kinase C α (PKCα), which supports angiogenesis through the induction of VEGF signaling [137]. The fenestrations showed a similar quantity and width to human primary GECs in vitro [115]. This group also demonstrated that the cytoskeleton modulators cytochalasin B and diamide can be utilized to further regulate fenestration quantity and structure. Furthermore, the hiPSC-derived ECs demonstrated size-selective permeability, with induced fenestration improving the passage of small solutes across a confluent monolayer of hiPSC-derived ECs while restricting the passage of large solutes across the same barrier.

Further improving upon the physiological relevance of hiPSC-derived models of EC fenestrations, Mou et al. were recently the first to discover the presence of fenestrations in hiPSC-derived ECs cultured in their silk-inspired glomerulus-on-a-chip system (Figure 3) [122]. This research group engineered an ultrathin biomimetic membrane to facilitate intercellular crosstalk between cocultured hiPSC-derived podocytes and ECs. This biomimetic membrane, fabricated from silk fibroin, also simplified the process of microtome sectioning for transmission electron microscopy analysis of fenestration formation. Mou et al. demonstrated that co-culture with podocytes supported fenestration formation that approached physiologically relevant values, covering approximately 40% of the EC area. Mou et al. also documented VEGF-A signaling between the podocytes and ECs in co-culture. This new microphysiological system opens up the possibility of addressing many questions that remain regarding the optimization of fenestration formation in hiPSC-derived ECs: (1) Do hiPSC-derived ECs cultured in microphysiological systems express diaphragms or arrange into organized sieve plates? (2) Does co-culture with podocytes or mesangial cells provide an advantage in physiological relevance over supplementation with VEGF-A or PMA alone? These hiPSC-derived microphysiological models also open the door to patient-specific disease models that can be used as drug-testing platforms for the restoration of kidney function by the regulation of fenestrations. Together, organ-on-a-chip microfluidic systems and hiPSC technologies provide new and promising avenues to engineer patient-specific models with definitive fenestrated endothelia for basic research and translational medicine, including mechanistic studies of EC fenestration development, structure, and function.

While separate microphysiological models demonstrating fenestrated endothelium have been designed for both the kidney glomerulus and proximal tubules, a combined kidney-on-a-chip model that supports fenestrated endothelia in both of these units of the nephron and integrates with renal vasculature has yet to be designed. There only a handful of published kidney-on-a-chip models to date combining the glomerulus, proximal tubules, and renal vasculature, none of which reports the presence of endothelial fenestrations [138,139,140,141,142]. Studying fenestration dynamics in a combined kidney-on-a-chip model would enhance understanding of the crosstalk between cell types found in the glomerulus and the proximal tubules such as glomerular endothelial cells and tubular epithelial cells [143]. These models would also increase knowledge of the differences and similarities between fenestrations found in these different units of the nephron.

In summary, hiPSC-derived microphysiological systems with EC fenestrations can support further investigation of the development of fenestrations for different cell types such as the GECs of the kidney glomerulus. Further, these models can reveal key mechanisms controlling the dynamics of fenestration structure and function in healthy and diseased states. Such investigations could elucidate novel targets and facilitate the development of therapeutics for kidney diseases that impact fenestration structure and function. Microphysiological systems provide the microenvironment to study the role of fenestrations in injury-mediated angiogenesis and vascular regeneration. These models can support the translation of information learned from pro-regenerative fenestrated endothelial models such as the liver, retina, and bone to the unique vasculature of the kidney for the inhibition of barrier degeneration and promotion of regeneration. Advancements in models of fenestrated endothelia across different organs will further improve our understanding of vascular perfusion and trans-endothelial transport across the human body.

## Figures and Tables

**Figure 1 ijms-25-09107-f001:**
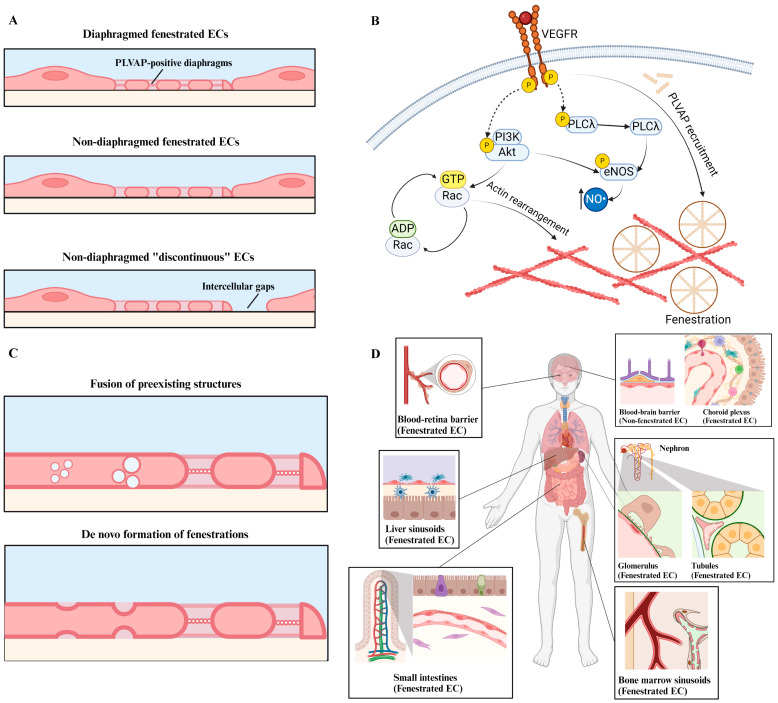
Schematic illustration of endothelial fenestration development and their distribution in various human organs. (**A**) Types of endothelial cell fenestrations, including diaphragmed fenestrations with PLVAP-positive diaphragms (top), non-diaphragmed fenestrations (middle), and non-diaphragmed “discontinuations” with intercellular gaps (bottom). Created with BioRender (https://app.biorender.com/, accessed on 22 July 2024). (**B**) Developmental pathways implicated in endothelial fenestrations and downstream of VEGF signaling. NO: Nitric oxide. Created with BioRender (https://app.biorender.com/, accessed on 22 July 2024). (**C**) Hypothesis of structural changes in endothelial cells during fenestration development. Created with BioRender. (**D**) Schematic of a human body highlighting various organs that contain fenestrated endothelial cells. Endothelial cells are labeled in pink. Created with BioRender (https://app.biorender.com/, accessed on 22 July 2024).

**Figure 2 ijms-25-09107-f002:**
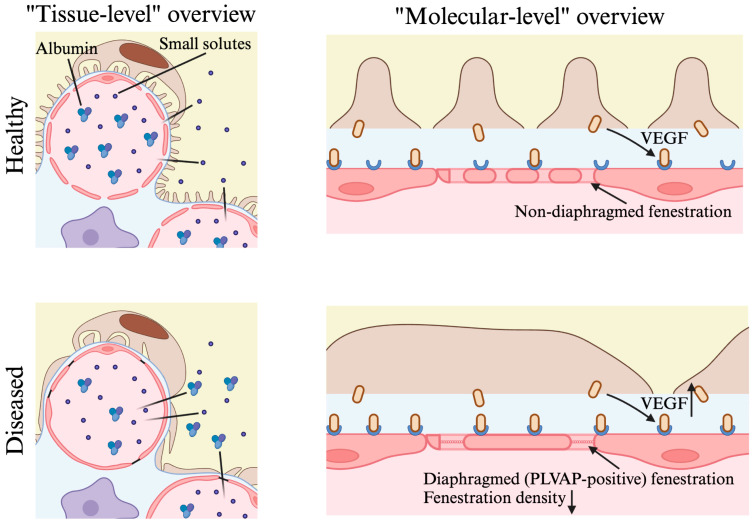
Illustration of glomerular endothelial cells in kidney glomerulus under healthy condition at the tissue level, with size-selective filtration function (**top left**) and at the molecular level with non-diaphragmed endothelial fenestrations (**top right**), compared to diseased condition at the tissue level with the loss of size-selective filtration function (**bottom left**) and at the molecular level with the presence of fenestration diaphragms (PLVAP-positive) along with increased VEGF signaling (**bottom right**). Glomerular endothelial cells are labeled in pink, podocytes are labeled in brown. Created with BioRender (https://app.biorender.com/, accessed on 14 May 2024).

**Figure 3 ijms-25-09107-f003:**
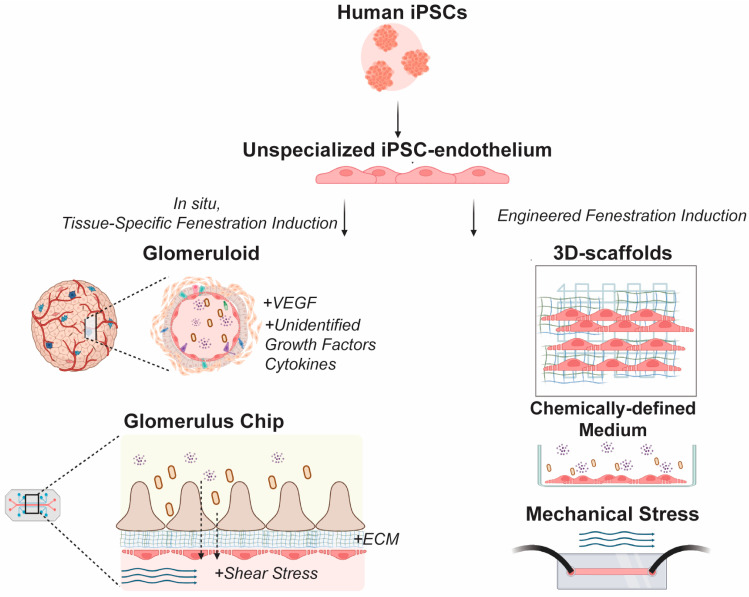
Engineered environments that induce tissue-specific fenestrations in ECs derived from unspecialized hiPSCs. Illustration of tissue-specific EC fenestration induction via in situ development including Glomeruloid, with influences by endogenous secretion of VEGF and other currently unidentified growth factors and cytokines (**top left**); Glomerulus Chip, with influences by VEGF, other growth factors, and cytokines, as well as mechanical factors (e.g., shear stress) and cell-laden ECM with physiologically-relevant structure (**bottom left**). Illustration of tissue-specific EC fenestration induction with defined variables including 3D-scaffolds (**top right**), chemically-defined medium (**middle right**), or mechanical stress (**bottom right**). Created with BioRender (https://app.biorender.com/, accessed on 13 June 2024).

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
