# Peer review of "Fenestrated Endothelial Cells across Organs: Insights into Kidney Function and Disease"

_ijms, 2024, doi:10.3390/ijms25169107_

Round 1
Reviewer 1 Report
Comments and Suggestions for Authors
General comment: The work presented by these authors arouses much scientific interest, bringing to light a highly interesting mechanism. The authors present this topic with precision and in detail, organizing the information collected in a schematic way. Below a few tips that could improve the work:
I suggest to reduce the introduction, that consist in well established knowledges.
2.1. This part is strongly interesting, I think that can be improved by a graphic explanation of the formation and dynamic of fenestrations.
Please reduce the repetition of information within the text.
line 178: If it is possible, I suggest to add some examples of the cited FDA- approved drugs.
Just out of my curiosity, I ask the authors how stable these structures are within the body, whether they remain dynamically spread in the different apparatuses throughout life or with aging and increasing endothelial senescence, atersclerosis etc these structures decrease?It would also be interesting to evaluate/discuss how the dynamism and the probable different percentage of fenestrations can affect the metabolism of drugs over life.
Author Response
Reviewer 1
Comments 1: General comment: The work presented by these authors arouses much scientific interest, bringing to light a highly interesting mechanism. The authors present this topic with precision and in detail, organizing the information collected in a schematic way. Below a few tips that could improve the work:
We thank the reviewer for their positive comments and underscoring the high scientific interest of our review manuscript..
Comments 2: I suggest to reduce the introduction, that consist in well established knowledges.
As suggested, we have reduced the content of the Introduction. Specifically, we removed contents from Page 1-Line 35 to Line 41, Page 2-Line 44 to Line 50, Line 61, Line 64 to Line 69.
Comments 3: 2.1. This part is strongly interesting, I think that can be improved by a graphic explanation of the formation and dynamic of fenestrations.
We thank the reviewer for their positive comment and helpful suggestion. In the revised version of the manuscript, we have added two additional graphical illustration panels in Fig. 1, (Fig. 1B and Fig. 1C) to illustrate the formation and dynamics of fenestrations.
Comments 4: Please reduce the repetition of information within the text.
We have reduced repetitive statements throughout the manuscript. Specifically, the changes include removal of Page 4-Line 154 to Line 158, Page 9-Line 408, Page 14-Line 622 to Line 623, Line 627 to Line 628, Line 630 to Line 635, Line 643 to Line 646, Page 16-Line 728, Page 19-Line 827 to Line 828.
Comments 5: line 178: If it is possible, I suggest to add some examples of the cited FDA- approved drugs.
Per the reviewer comment, we have added examples of FDA-approved VEGF-targeting cancer drugs. This revision can be found on Page 5-Line 186 to Line 187.
Comments 6: Just out of my curiosity, I ask the authors how stable these structures are within the body, whether they remain dynamically spread in the different apparatuses throughout life or with aging and increasing endothelial senescence, atersclerosis etc these structures decrease?It would also be interesting to evaluate/discuss how the dynamism and the probable different percentage of fenestrations can affect the metabolism of drugs over life.
We thank the reviewer for their intellectual curiosity and insightful comments. In the revised version of this review manuscript, we have added additional information regarding dynamic changes of endothelial fenestrations and provided specific examples in the liver. We also discussed how change in fenestrations can influence or alter drug metabolism (Page 7-Line 298 to Page 8-Line 320). We have also added a brief discussion of how decline in glomerular filtration rate in aging can potentially be attributed to changes in endothelial cell fenestrations. Notably, we also acknowledge that there’s limited research on this topic (Page 11-Line 469 to Line 476), making it a potential future study to help advance the field.
Reviewer 2 Report
Comments and Suggestions for Authors
This manuscript, although extensive provide very few novel information. It is merely textbook material. The novelty is in some details of molecular development and organization of the fenestrated endothelial cells. Introduction part in relation to distribution of fenestrated endothelial cells may be considered not necessary, since the rest of the manuscript is focused on kidney's physiology and pathophysiology.
I have no specific comments, since manuscript is clearly written, however, it would benefit from focusing on kidneys and providing more molecular details. Also, novelty of materials should be emphasized.
Author Response
Reviewer 2
Comments 1: This manuscript, although extensive provide very few novel information. It is merely textbook material. The novelty is in some details of molecular development and organization of the fenestrated endothelial cells. Introduction part in relation to distribution of fenestrated endothelial cells may be considered not necessary, since the rest of the manuscript is focused on kidney's physiology and pathophysiology.
Because this is a review article for a broad community of readers and scientists who may or may not be experts on this specific topic, we believe it is important to briefly introduce where in the body different types of fenestrated endothelial cells can be located. Such discussion is also appropriate for the Introduction section, as we discuss in detail fenestrated endothelial cells in brain, GI tract, liver, bone, and retina, before diving into kidneys. We ensured that the title of the manuscript appropriately reflects the content of the manuscript, starting broadly and then focusing more of the kidneys.
Comments 2: I have no specific comments, since manuscript is clearly written, however, it would benefit from focusing on kidneys and providing more molecular details. Also, novelty of materials should be emphasized.
We have added molecular details on current knowledge of glomerular endothelial fenestration development and dynamics (Page 9-Line 412 to Page 10-Line 433).
We made multiple revisions to help emphasize new insights in this review manuscript . Below is a summary of the additional changes made to help address this comment.
- We added additional content to the Introduction section (Page 2-Line 78 to Line 83) to provide an overview of the current state of the field, knowledge gaps, and how emerging technologies including microphysiological systems could be harnessed to advance current understanding of the field and related topics
- We added a brief discussion of how natural decline in glomerular filtration in aging kidneys could potentially be attributed partly to changes in endothelial fenestrations, which merits further research (Page 11-Line 469 to Line 476)
- We discussed how microtome sectioning techniques or membrane materials could enhance the feasibility of using microphysiological models to study fenestrations (Page 17-Line 755 to Line 762)
- We included an oversiew and discussions of the knowledge gaps in microphysiological models of fenestration dynamics in kidney disease (Page 17-Line 780 to Line 785)
- We highlighted the potential application of hiPSC-derived microphysiological models for patient-specific drug testing platforms to examine fenestration dynamics in kidney disease (Page 19-Line 835 to Line 837)
- Contributed new ideas for potential future studies. For example, we suggested future studies to create a kidney-on-a-chip platform that combines the kidney glomerulus, proximal tubules, and renal vasculature and supports endothelial fenestrations to improve physiological relevance and disease modeling capabilities (Page 19-Line 842 to Line 854)
Additional revisions
- In the returned peer reviewed version of the manuscript and template, we noticed an error in subtitles of 2.1. and 2.2. This is now corrected by swapping the subtitles to the correct locations.
- We encountered some Microsoft Word software bug, leading to some text getting replaced by [OBJ]. Therefore, we added our original text back wherever applicable.
Round 2
Reviewer 2 Report
Comments and Suggestions for Authors
AUthors provided revised version of their manuscript on the development, distribution and physiological and pathological significance of fenestrated endothelial cells. However, the manuscript still contains a vast amount of textbook materials and is still lacking the molecular basis (which is the section that the manuscript is sent to). For example, the first sentence in Introduction: "The vascular (or circulatory) system is composed of vessels that carry blood, oxygen, and essential nutrients to the body’s tissues and organs." is classic textbook introduction sentence and should be omitted. There are many like that throughout the manuscript. This kind of information are generally accepted and known and manuscript, albeit review, should provide novel approach and novel, current information on the state of the topic in the field.
Further, manuscript will benefit from some table or additional figures comparing different organs' EC in regard of their specificity.
Novelties should be emphasized.
Author Response
Authors provided revised version of their manuscript on the development, distribution and physiological and pathological significance of fenestrated endothelial cells. However, the manuscript still contains a vast amount of textbook materials and is still lacking the molecular basis (which is the section that the manuscript is sent to). For example, the first sentence in Introduction: "The vascular (or circulatory) system is composed of vessels that carry blood, oxygen, and essential nutrients to the body’s tissues and organs." is classic textbook introduction sentence and should be omitted. There are many like that throughout the manuscript. This kind of information are generally accepted and known and manuscript, albeit review, should provide novel approach and novel, current information on the state of the topic in the field.
We thank the reviewer for their suggestions. We have removed or revised content that experts may suggest is textbook-like information. We made these revisions such that the clarity of information and relevance to multidisciplinary audience is retained or not compromised. This review article is appropriate for facilitating cross-discipline learning and collaboration. With most of our cited references spanning the past 5 years, this review article addresses the most current state of the field in EC fenestration not covered fully by prior work.
To address the reviewer’s suggestion to minimize background information that may seem like common knowledge, we have deleted content including: the first 2 sentences of the introduction (as suggested) (Page 1), first paragraph and the last 2 sentences of the Brain section (Page 6), the first 5 sentences of the Gastrointestinal tract section (Page 7), the first sentence of the Liver section (Page 7), and the first sentence of the Bone section (Page 8). We also rephrased the first sentence of the Kidneys section (Page 10), and we removed the first sentence introducing renal tubules (Page 11).
To provide more insights into the molecular basis of endothelial fenestration, we added
1) information demonstrating molecular roles of various VEGF subtypes and Ccbe1 in choroid plexus EC development (Page 6 Line 193 – 199).
2) additional information regarding PLVAP expression changes in intestinal EC fenestration under bacteria infection, as an extension of intestinal EC’s role in immune response (Page 7 Line 223 – 227).
3) more details regarding GEC fenestration regulation (Page 15 Line 592 – 594).
4) the significance of using microphysiological systems in modeling and studying EC fenestrations, supported by shear-stress-induced mechanosensitive pathways that can affect EC fenestration (Page 15 Line 622 – Page 16 Line 640),
5) discussion of the issues with reliance on rodent models as motivated by species-specific differences in VEGF signaling (Page 15 Line 607 – 612) and differences in Six1/Six2 regulation (Page 19 Line 761 – 765).
Additionally, in our 1st round revised manuscript, we provided various molecular insights spanning from EC fenestration developmental pathways (Page 3 Line 93 to Page 4 Line 151), to differential protease regulation in unique intestinal EC fenestration formation (Page 7 Line 206 – 216), to LSEC molecular properties and how aging affect their molecular response in liver metabolism (Page 7 Line 249 – 288). Also, we have provided molecular basis of bone marrow sinusoidal EC homeostasis and disease (Page 8 Line 294 – 332), and the detailed role of PLVAP in regulating retina EC function how pathological changes in retina EC can be associated with various diseases (Page 9 Line 341 – 359). Moreover, we have provided detailed molecular pathways involved in GEC fenestration development (Page 10 Line 376 – 403). In the kidney disease section, we have provided discussion on the role of VEGF and EHD3 in DN and how they can be utilized to tackle DN (Page 12 Line 452 to Page 13 Line 494), while we also discussed the role of PLVAP in FSGS and AS (Page 13 Line 498 – 512 and Page 13 Line 520 – 523). Additionally, we have discussed how various molecules are aberrantly regulated in preeclampsia and how these molecular targets can be utilized to treat preeclampsia (Page 14 Line 538 – 551). In the Future Perspectives section, we have discussed how EHD3 can be utilized to treat DN (Page 14 Line 600 – 625), and how hiPSC-derived ECs can be induced to express fenestrations in vitro using various molecular cues (Page 14 Line 573 to Page 15 Line 592).
Together, the content of this review article provides appropriate background information accompanied by insightful discussions of the most current state of the field. Thus, this review article is appropriate for both new and experienced researchers, scientists, and students who may be interested in the topic.
Further, manuscript will benefit from some table or additional figures comparing different organs' EC in regard of their specificity.
We thank the reviewer for this suggestion. In the revised version of the manuscript, we have added a Table (Table 1) summarizing fenestrated ECs in various organs and their characteristics and function. Specifically, the table highlights the structure and function of EC fenestrations (e.g. diaphragmed, basement membrane, (dis)continuous), such that the reader can quickly define the commonalities and distinctions between the highlighted organs/tissues.
Novelties should be emphasized.
We have included the most up-to-date literature (many from the most recent 5 years) studying the roles of EC fenestration in healthy and diseased conditions in various organs. We also discussed how the cutting-edge microphysiological system technologies and hiPSC differentiation technologies can be utilized to model and study EC fenestrations, with most recent literatures being discussed throughout this section – including papers published just days before our manuscript submission, and therefore unlikely to have been reported by past review articles or textbooks in the field. Additionally, we discussed pressing/unknown questions in the field, providing insights for future work.
Should the reviewer have additional concerns about our topic (which was previously approved by the editors), we would appreciate specific examples they may have of review articles they believe cover all the topics and insights presented in our submitted manuscript. This will help us address their comment fully and better delineate our work from others.
Round 3
Reviewer 2 Report
Comments and Suggestions for Authors Manuscript is improved. I think that it can be accepted in this form for publication. There is a minor objection- all accronyms should be defined with full words when appearing the first time (e.g. first sentence of the first chapter, in Introduction, but there is more in a few places).Manuscript is improved. I think that it can be accepted in this form for publication. There is a minor objection- all accronyms should be defined with full words when appearing the first time (e.g. first sentence of the first chapter, in Introduction, but there is more in a few places). This can be done during editing the proofs.